# Glacier extraction based on high-spatial-resolution remote-sensing images using a deep-learning approach with attention mechanism

**Xinde Chu**[1], **Xiaojun Yao**[1], **Hongyu Duan**[1], **Cong Chen**[2], **Jing Li**[1], **and Wenlong Pang**[3]

[1]College of Geography and Environmental Science, Northwest Normal University, Lanzhou, 730070, China
[2]Key Laboratory of Western China's Environmental Systems (Ministry of Education), College of Earth and Environmental Sciences, Lanzhou University, Lanzhou, 730000, China
[3]Xining Center of Natural Resources Comprehensive Survey, China Geological Survey, Xining, 810000, China

**Correspondence:** Xiaojun Yao (xj_yao@nwnu.edu.cn)

**Abstract.** The accurate and rapid extraction of glacier boundaries plays an important role in the study of glacier inventory, glacier change and glacier movement. With the successive launches of high-resolution remote-sensing satellites and the increasing abundance of available remote-sensing data, great opportunities and challenges now exist. In this study, we improved the DeepLab V3+ as Attention DeepLab V3+ and designed a complete solution based on the improved network to automatically extract glacier outlines from Gaofen-6 panchromatic and multispectral (PMS) images with a spatial resolution of 2 m. In the solution, test-time augmentation (TTA) was adopted to increase model robustness, and the convolutional block attention module (CBAM) was added into the atrous spatial pyramid pooling (ASPP) structure in DeepLab V3+ to enhance the weight of the target pixels and reduce the impact of superfluous features. The results show that the improved model effectively increases the robustness of the model, enhances the weight of target image elements and reduces the influence of non-target elements. Compared with deep-learning models, such as full convolutional network (FCN), U-Net and DeepLab V3+, the improved model performs better in the test dataset. Moreover, our method achieves superior performance for glacier boundary extraction in parts of the Tanggula Mountains, the Kunlun Mountains and the Qilian Mountains based on Gaofen-6 PMS images. It could distinguish glaciers from terminal moraine lakes, thin snow and clouds, thus demonstrating excellent performance and great potential for rapid and precise extraction of glacier boundaries.

## 1 Introduction

The cryosphere is one of the five major circles of climate systems (Li et al., 2008), of which mountain glaciers are an important part. Changes in mountain glaciers are closely related to regional climate and are regarded as natural indicators and early warnings of climate change (Oerlemans, 1994; Pfeffer et al., 2008; Azam et al., 2018). Since the second half of the 20th century, climate warming has led to rapid shrinkage of glaciers globally (Yao et al., 2012), producing major impacts on the utilization of regional water resources and rising sea levels (King et al., 2012; Grinsted, 2013; Schrama et al., 2014). The accurate extraction of glacier boundaries can assist to detect the status of glacier areas (Racoviteanu et al., 2015) and elucidate the response pattern of glaciers to climate change (Bishop et al., 2004; Sun et al., 2018). As a large-range and long-range sensing technology for ground exploration, remote sensing can rapidly obtain glacier information, including its boundary, velocity, etc. (Robson et al., 2015; Zhang et al., 2019). However, most extant research on glacier changes and glacier inventories is based on remote-sensing images of low to medium resolution, such as the Landsat series and Aster images (Liu et al., 2020; Zhao et al., 2020), which may result in an inaccurate estimation of the global glacier resource to some extent (e.g., the glacier area threshold for the Randolph Glacier Inventory (RGI) is $0.01\,\mathrm{km}^2$). Available remote-sensing data are increasingly abundant due to successive launches of high-resolution remote-sensing satellites. Indeed, the efficient and rapid acquisition of glacier bound-

aries based on these data currently constitutes a frontier issue in glacier remote-sensing research.

Glacier boundaries are generally extracted by manual visual interpretation and semi-automatic or automatic methods. The former can yield relatively accurate results, but applications based on high-resolution imagery over a wide area are both time-consuming and laborious (Yan and Wang, 2013). The latter extracts glaciers based on their spectral differences from other features; e.g., snow and ice have a strong absorption in the shortwave infrared band (1.55–1.75 µm) and a robust reflection in the visible to near-infrared band (0.45–0.90 µm) (Guo et al., 2017). Methods used in this approach mainly include the ratio method (Ji et al., 2020), the snow cover index (Y. L. Wang et al., 2021), and supervised classification and unsupervised classification (Nie et al., 2010). However, the absence of a shortwave infrared band in some high-resolution optical remote-sensing images (e.g., Quick-Bird satellite images, WorldView-2 satellite images, SPOT-6 NAOMI, and Gaofen-6 panchromatic and multispectral – PMS) limits the application of the ratio method and normalized difference snow index (NDSI), which have a better extraction effect.

Deep learning has been widely adopted in the field of computer vision and image processing in recent years (Girshick, 2015). It can automatically obtain mid- and high-level abstract features from images due to its powerful feature learning and characterization capabilities compared with traditional classification methods (Redmon et al., 2016). At present, numerous typical deep-learning models, such as full convolutional network (FCN) (Long et al., 2015), Segnet (Badrinarayanan et al., 2017), U-Net (Ronneberger et al., 2015) and the DeepLab series (Chen et al., 2018), have been successfully applied to the semantic segmentation task of remote-sensing images (Huang et al., 2018; Tong et al., 2020), including the cryosphere domain. For example, Zhang et al. (2019) automatically delineated the calving front of the Jakobshavn Isbræ Glacier using a deep-learning method; Robson et al. (2020) combined deep-learning and object-based image analysis to extract rock glacier boundaries in the La Laguna catchment in northern Chile and the Poiqu catchment in the Central Himalaya; He et al. (2021) extracted the glacial lakes of the Alatau mountains of Tianshan through deep learning; Marochov et al. (2021) segmented Sentinel-2 image-covered marine-terminating outlet glaciers in Greenland into seven classes using the convolutional neural network (CNN); Baumhoer et al. (2019) extracted Antarctic glacier and ice shelf fronts at nine locations using deep learning based on Sentinel-1 images; and Xie et al. (2020) built the GlacierNet for debris-covered glacier mapping. However, current methods may not make optimal use of glacier features, and most studies are limited to extracting small-scale glaciers in some areas using low- to medium-resolution remote-sensing images, which may overlook some smaller glaciers. Therefore, the combination of deep learning and attention mechanism has the potential to provide an effective and powerful technique for automatic extraction of mountain glaciers.

The main objective of this study is to propose a new method for automatic extraction of glacier boundaries from high-resolution Gaofen-6 PMS images based on the DeepLab V3+ network and attention mechanism. Then, the accuracy and robustness of the proposed method are ascertained through a comparison with the reference outlines of glaciers based on the manual interpretation of orthorectified images, taking parts of the Tanggula Mountains, Kunlun Mountains and Qilian Mountains as the test region. Simultaneously, we assess our result by comparing it with the GAM-DAM (Glacier Area Mapping for Discharge from the Asian Mountains) glacier inventory (GGI) (Nuimura et al., 2015) and the glacier coverage data on the Tibetan Plateau in 2017 (TPG2017) (Ye et al., 2017; Ye, 2019).

## 2 Model structure and data process scheme

In this study, we used DeepLab V3+ in combination with the attention mechanism (Sect. 2.1) to explore the glacier extraction method based on high-spatial-resolution images. The proposed framework for using deep learning to extract glaciers is summarized in Fig. 1. The Gaofen-6 images were preprocessed and divided into two groups. Firstly, the former was used to make samples by data augmentation to train the improved DeepLab V3+, and then the latter was used to test the trained model by performing the classification (Sect. 2.3). Meanwhile, test-time augmentation (TTA) was added to this classification to improve model accuracy, which is described in Sect. 2.4.

### 2.1 Network structure

DeepLab V3+, an encoding–decoding architecture proposed by Chen et al. (2018), is currently one of the most advanced semantic segmentation algorithms. In this paper, we improved the DeepLab V3+ model by adding an attention mechanism to the encoding–decoding structure (Attention DeepLab V3+). In the encoder, ResNet-34 (He et al., 2016) was used as the backbone network (Sect. 2.1.3) to extract semantic information to obtain low-level features, and then the atrous spatial pyramid pooling (ASPP) module with attention mechanism was connected to obtain the encoder feature. ASPP is a parallel structure with dilated convolution, which expands the perceptual field to obtain multi-scale contextual information from the image. The attention mechanism, on the other hand, increases the weight of the target image element. These two parts will be described in detail in Sect. 2.1.1 and 2.1.2, respectively.

In the decoder, the low-level features and the encoder feature maps were input. The encoder feature performed upsampling with a factor of 4 and then fused with the low-level features. Subsequently, a depthwise separable convolution

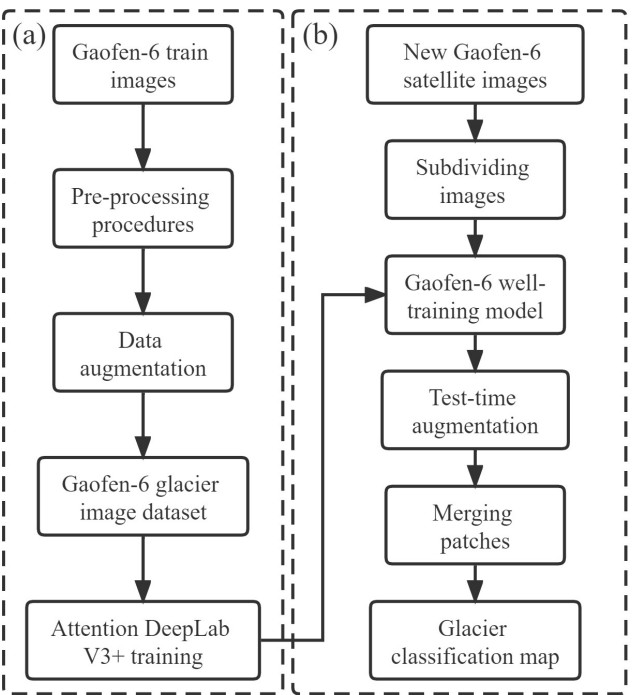

**Figure 1.** Overall flow of glacier extraction based on deep learning. **(a)** The model training. **(b)** The prediction process.

(Sect. 2.1.4) and a bilinear interpolation upsampling with a factor of 4 were executed to extract the expected features and output them at the same size as the input image (Fig. 2).

### 2.1.1 Attention mechanism

The attention mechanism can learn contextual information and capture internal correlations. Its basic idea is to ignore irrelevant information and focus on key information in operations (Woo et al., 2018). In this paper, we used the convolutional block attention module (CBAM) (Fig. 3), including a channel attention module (CAM) and a spatial attention module (SAM), to obtain the channel attention weights ($F_C$) and spatial attention weights ($F_S$) in turn. We then multiplied these by the original feature map $A \in R^{C \times H \times W}$ to adaptively adjust the features and increase the weights of target features to generate refined feature $F \in R^{C \times H \times W}$ (Woo et al., 2018):

$$F = A \otimes F_C \otimes F_S , \tag{1}$$

where $\otimes$ stands for the element-wise multiplication. If the two operands have different dimensions, the values are broadcast (copied) in such a way that the spatial attention values are broadcast along the channel dimension and that the channel attention values are broadcast along the spatial dimension.

In CAM, average-pooling and max-pooling operations were used to aggregate spatial information of a feature map, generating two different spatial context descriptors: $F_{cavg}$ and $F_{cmax}$. Both are then forwarded to a shared multi-layer

perceptron (MLP) to generate the output features, which are then merged using element-wise summation. The merged sum is finally sent to the sigmoid function $\sigma$ to produce the channel attention map $F_C \in R^{C \times 1 \times 1}$. To reduce the parameter resources, the hidden size of MLP is set to $R^{C/r \times 1 \times 1}$, where $r$ is defined as the reduction ratio. The channel attention is computed as follows:

$$F_C = \sigma \left( W_1 \left( W_0 \left( F_{cavg} \right) \right) + W_1 \left( W_0 \left( F_{cmax} \right) \right) \right) , \tag{2}$$

where $W_0 \in R^{C/r \times C}$ and $W_1 \in R^{C \times C/r}$ stand for the MLP weights, which are shared for both inputs, and the rectified linear unit (ReLU) activation function is followed by $W_0$.

In SAM, average-pooling and max-pooling operations were used to aggregate channel information to generate two 2D maps: $F_{savg} \in R^{1 \times H \times W}$ and $F_{smax} \in R^{1 \times H \times W}$. Each denotes average-pooled features and max-pooled features across the channel, respectively. They are then concatenated and convolved by a standard convolution layer, producing the 2D spatial attention map $F_S \in R^{1 \times H \times W}$. The spatial attention is computed as follows:

$$F_S = \sigma ( f^{7 \times 7}([F_{savg}; F_{smax}])) , \tag{3}$$

where $\sigma$ is the sigmoid function, and $f^{7 \times 7}$ represents a convolution operation with the filter size of $7 \times 7$.

### 2.1.2 Attention ASPP

Dilated/atrous convolution allows us to explicitly control the resolution of features computed by deep convolutional neural networks and adjust the filter's field-of-view in order to capture multi-scale information. Dilated convolution has an additional hyper-parameter, termed the dilation rate, which refers to the number of intervals in the kernel (e.g., the normal convolution is a dilatation rate of 1). The ASPP structure replaces the partial volumes of the deep neural network with the dilated convolution (Yu and Koltun, 2015), which expands the perceptual field without increasing the parameters, thus obtaining more feature information. The structure consists of a $1 \times 1$ convolutional layer, a pooling layer and three dilated convolutions with expansion rates of 6, 12 and 18 in parallel.

However, the dilated convolution may cause discontinuity of spatial information, which was addressed by incorporating CBAM into the juxtaposition structure of the extracted features in this paper. The role of the attention mechanism was to focus on the noticed target pixel and enhance its weight, and the dilated convolution in the ASPP can obtain contextual information at different scales. Adding the attention mechanism could make the features of related categories more aggregated, thus effectively reducing the phenomenon of empty features. The five branches of the juxtaposition structure obtained more contextual information by collecting features from different sensory domains, and, by combining this information with refined feature maps, dependencies be-

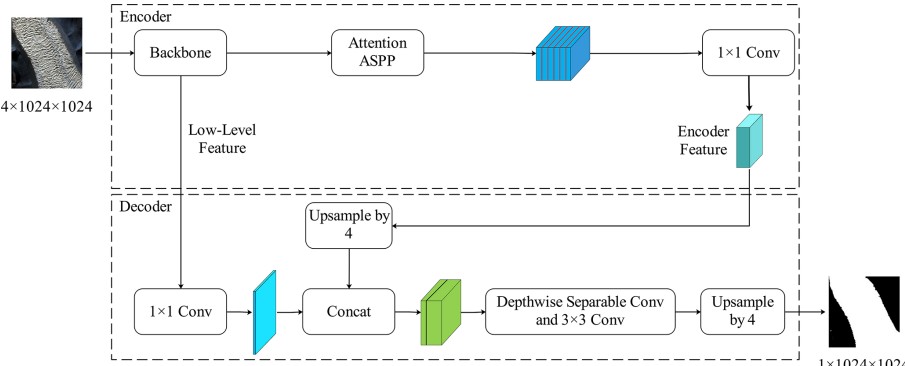

**Figure 2.** Architecture of Attention DeepLab V3+.

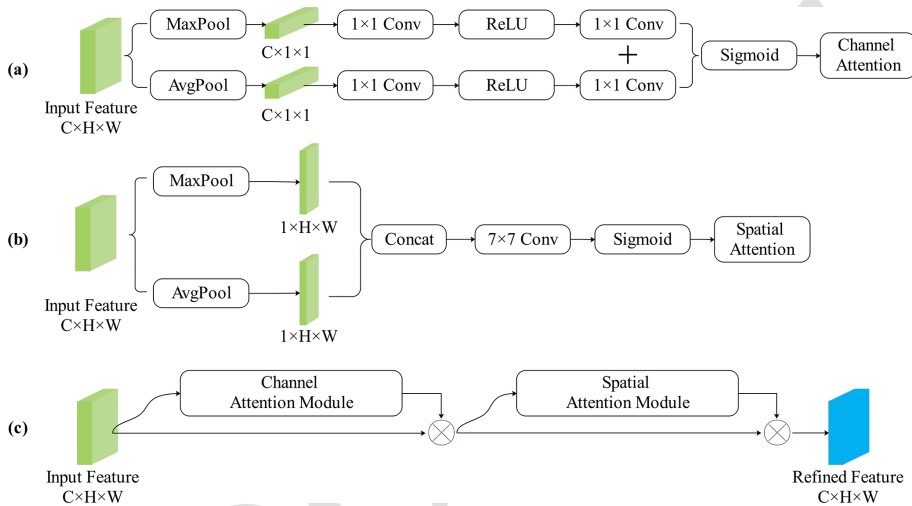

**Figure 3.** Attention module. **(a–c)** Channel attention module, spatial attention module and CBAM, respectively.

tween pixels and differences between categories were determined (Fig. 4).

### 2.1.3   Backbone

Although the ability of CNN to retrieve relevant information from images is enhanced with the increase in network depth (Telgarsky, 2016), a network that is too deep could lead to gradient explosion and network degradation. Residual connections (He et al., 2016) solved this problem by feeding a given layer into the previous one, where a building block of residual learning was included (Fig. 5) and by which the depth of the network and learning capacity can be dramatically increased. There are five versions of the ResNet model, with 18, 34, 50, 101 and 152 layers. To assess the trade-off between performance and computational efficiency, we used ResNet-34 as the backbone in our work. ResNet-34 consists of 16 blocks (Fig. 5) and 33 convolutional layers in total (He et al., 2016). The first convolutional layer of the overall model is followed by a max pooling layer. An average pooling layer, a full-connected layer and a softmax layer are

subsequent to the last convolutional layer. It is worth noting that the default input channel of ResNet-34 is 3. To classify images that have R, G, B and NIR bands, we adjusted the input size of the conversional ResNet-34 to four channels.

### 2.1.4   Depthwise separable convolution

In this paper, we added a depthwise separable convolution at the end of semantic segmentation after combining high-level and low-level features (Chollet, 2017). This convolution can decompose the traditional convolution into a depthwise convolution and a pointwise convolution.

Regarding depthwise convolution, one convolution kernel is responsible for one channel, and one channel is convolved by only one convolution kernel. Moreover, the number of feature map channels generated in the process is exactly the same as the number of input channels. This operation does not effectively utilize the feature information of different channels at the same spatial location. Therefore, pointwise convolution is needed to combine these feature maps to generate new feature maps (Howard et al., 2017). The operation

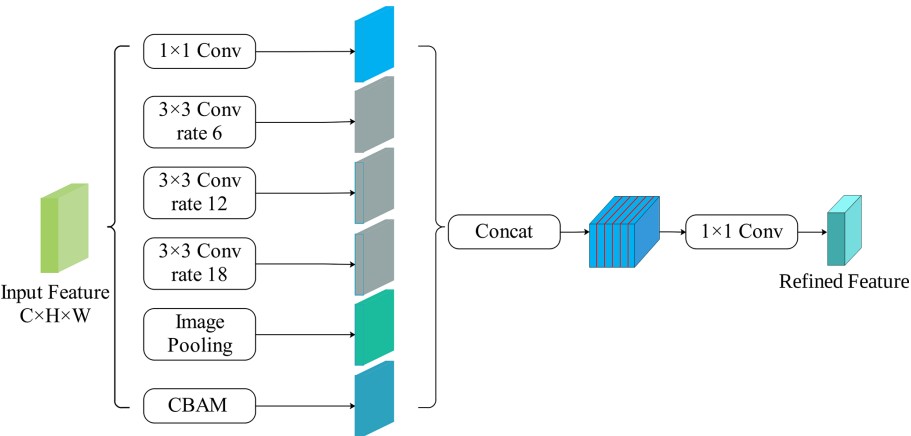

**Figure 4.** Structure of Attention ASPP.

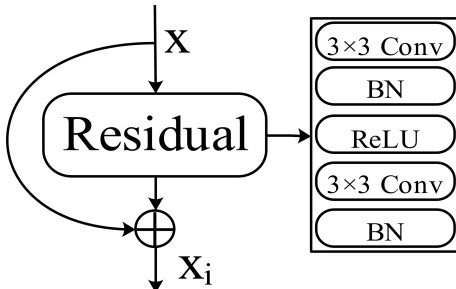

**Figure 5.** Residual connection. Each residual block is composed of convolutions (Conv), batch normalizations (BN), and rectified linear units (ReLU).

of pointwise convolution is highly similar to the conventional convolution operation in that the size of its convolution kernel is $M \times 1 \times 1$, and $M$ is the number of channels in the previous layer. The number of output feature maps is determined by the number of convolution kernels.

Depthwise separable convolution can improve the efficiency of operation without losing too much accuracy compared with normal convolution. In addition, the layers of the neural network with depthwise separable convolution can be deeper for the same number of parameters.

### 2.1.5 Loss function

Due to the high resolution of Gaofen-6 PMS images and the large size of some glaciers, only a small part of glaciers or other features may be present in the sample, which has the potential to cause sample imbalance. In this paper, dice loss (Milletari et al., 2016) was used as the loss function to mitigate this phenomenon. Dice loss is expressed as follows:

$$L_{\mathrm{d}} = 1 - \frac{I + \varepsilon}{U + \varepsilon} , \quad (4)$$

where $I$ is the number of intersections between sample labels and predicted pixels, $U$ is the sum of sample labels and predicted pixels, and $\varepsilon$ is a constant that is mainly used to prevent the denominator from being zero and smooth the loss operation.

The use of dice loss generally produces severe oscillations during the training of the model when the positive sample is a small target because a large loss change and a consequent gradient drastic change will occur once some pixels of the small target are predicted incorrectly in the case of only foreground and background. Therefore, we introduced a combination of cross-entropy loss and dice loss to make the network training more stable. The cross-entropy loss is as follows:

$$L_{\mathrm{c}} = \frac{1}{N} \sum_i -\left[ y_i \times \log(p_i) + (1 - y_i) \times \log(1 - p_i) \right] , \quad (5)$$

where $p_i$ is the probability that the pixel is predicted as a glacier, and $y_i$ is the sample label which takes the value of 1 if the sample pixel is a glacier and 0 otherwise.

Therefore, these above two losses were fused as the final loss function:

$$L = 0.5 \times (L_{\mathrm{c}} + L_{\mathrm{d}}) . \quad (6)$$

### 2.2 Preprocessing procedures of Gaofen-6 PMS image and datasets production

Deep learning requires a large amount of labeled data related to the classification target for training the model. However, currently available open-source datasets cannot meet the requirements of the classification in this paper. For this reason, we collected Gaofen-6 PMS images that were less cloudy and snowy from the China High-resolution Earth Observation System (https://www.cheosgrid.org.cn/, last access: 16 May 2022) (Table S1 in the Supplement) as a training and validation dataset of the model, some of which

were selected to test the accuracy of the glacier extraction method. The Gaofen-6 satellite, officially operational since 21 March 2019, is a low-orbiting optical remote-sensing satellite with high spatial resolution, featuring wide-coverage, high-quality and efficient imaging. A 2 m panchromatic and 8 m multispectral high-resolution camera and a 16 m multispectral medium-resolution wide-field camera are on board the Gaofen-6 satellite, the former with an observation width of 90 km and the latter with that of 800 km.

Preprocessing, including fusion, orthorectification, geometric alignment and some other operations, was performed prior to using the original defective images. Glaciers in the images were extracted as true values by manual visual interpretation. Considering the high spatial resolution of Gaofen-6 PMS images, the larger scale of glaciers and other features, and the inclusion of two kinds of features (glaciers and other features) in the samples whenever possible, the images were cropped to $1024 \times 1024$ size and used as the input for deep-learning training. It is important to prepare a sufficiently diverse dataset to ensure that the model can be adapted to different scenarios of glacier extraction. In this paper, data augmentation, including randomly clipped Gaofen-6 PMS images, vertical flipping, horizontal flipping and diagonal flipping of the samples, as well as clockwise $90°$ and counterclockwise $90°$ rotations, were conducted to expand the sample library, improve the model accuracy and enhance its generalization performance. Finally, a training set and a validation set containing 3600 well-annotated images of $1024 \times 1024$ size with blue, green, red and near-infrared bands were obtained. Meanwhile, a test set was kept containing 400 images without data augmentation. An example of the sample is shown in Fig. 6.

## 2.3 Complete glacier extraction

Since a sample usually displays only a portion of the glacier, an image larger than the sample size needs to be input to extract the complete glacier. In most cases, the images to be classified are generally split into a series of images with the same size as the samples and fed into the network for prediction, and then the predicted results are merged into one final result image in the cropping order in the prediction process. However, larger classification errors and unsmooth merging after clipping could occur due to the insufficient pixel features in the edge areas of each patch. Consequently, we adopt the strategy of making the two clipped patches overlap each other to preserve the features of edge pixels and make the merging of edges smoother.

Figure 7 illustrates the process of prediction, in which the preprocessed original image over the area in the red checkbox (Fig. 7a and b) must be split into $1024 \times 1024$ overlapping patches prior to being input into the network for prediction, and then the classification results of each patch are obtained (Fig. 7d). Subsequently, 90 % of the central part of each patch (red checkbox in Fig. 7d) was merged to obtain the classification result (Fig. 7c).

## 2.4 Test-time augmentation

In this paper, the test-time augmentation (TTA) strategy was used in result extraction and can be considered as a postprocessing technique because it is executed during the testing phase (Z. Q. Wang et al., 2021). Therefore, it does not affect the network learning parameters but rather attempts to obtain multiple enhanced copies by performing data enhancement operations, such as horizontal and vertical transformations, on each image during the test and then combines the results of multiple enhanced copies for prediction (Fig. 8). The voting formula is as follows:

$$p = \frac{\sum_{v=1}^{n} S}{n} \, , \tag{7}$$

where $p$ is the probability that the pixel belongs to a glacier, $n$ is the number of each test image and its copies, and $S$ is the probability of each pixel belonging to the glacier in the image and its copies. Figure 8 presents the result of TTA.

## 2.5 Evaluation metrics

To quantitatively describe the ability to extract glaciers from high-resolution images using the method proposed in this study, the results obtained from manual visual interpretation were chosen as ground truth to be compared with the classification results. We selected the kappa, mean intersection over union (MIoU), $F1$ score ($F1$) and average symmetric surface distance (ASD) as the metrics. The kappa, MIoU and $F1$ range from 0 to 1, with 1 being equivalent to 100 % accuracy.

The kappa is a statistic that measures the agreement between prediction and ground truth (Olofsson et al., 2014) and can be calculated as follows:

$$p_e = \frac{(TP + FP) \times (TP + FN) + (FN + TN) \times (FP + TN)}{n^2} \, , \tag{8}$$

$$\text{kappa} = \frac{\frac{(TP+TN)}{n} - p_e}{1 - p_e} \, , \tag{9}$$

where $n$ is the number of total pixels, TP and TN are the number of pixels correctly predicted to be glaciers and correctly predicted to be other features, respectively, and FP and FN are the number of pixels incorrectly predicted as glaciers and incorrectly predicted as other features, respectively.

MIoU for the binary classification problem (He et al., 2021) is as follows:

$$\text{MIoU} = \frac{\frac{TP}{FP+TP+FN} + \frac{TN}{FP+TN+FN}}{2} \, . \tag{10}$$

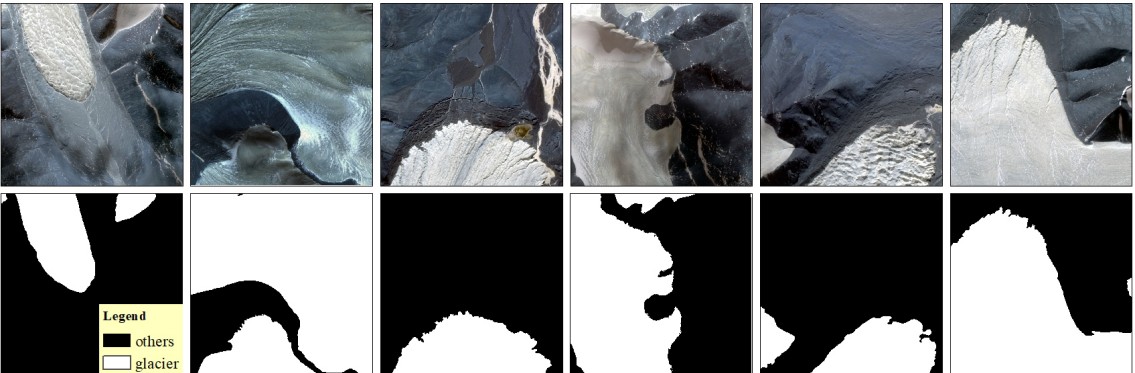

**Figure 6.** The RGB Gaofen-6 samples and ground truth are displayed in the first and second rows, respectively.

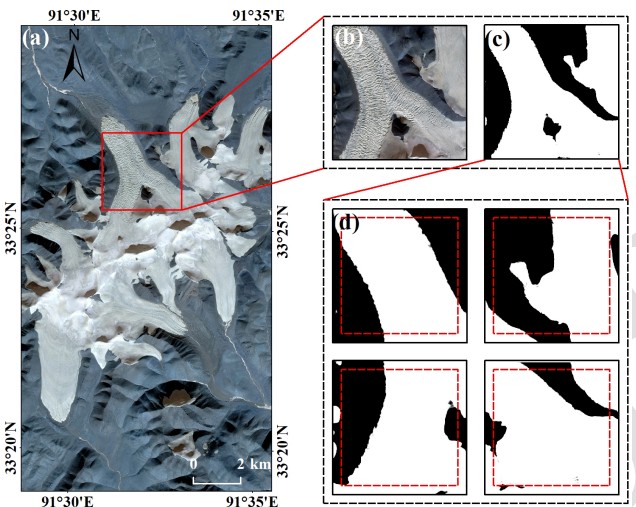

**Figure 7.** Postprocessing. Panels **(a)** and **(b)** are the RGB images obtained on 20 December 2020, **(c)** the final result of merging, and **(d)** the prediction results.

$F1$ is the harmonic average of precision rate ($P$) and recall rate ($R$) (Marochov et al., 2021):

$$P = \frac{\text{TP}}{\text{TP} + \text{FP}} , \tag{11}$$

$$R = \frac{\text{TP}}{\text{TP} + \text{FN}} , \tag{12}$$

$$F1 = 2 \times \frac{P \times R}{P + R} . \tag{13}$$

ASD is given in pixels and meters and is based on the boundaries of the prediction result and ground truth (gt). For each pixel of a boundary in the prediction result, the Euclidean distance to the closest pixel of another boundary in gt is calculated using the approximate nearest-neighbor technique and stored. In order to provide symmetry, the same process is applied from the boundary of gt to the prediction result (Heimann et al., 2009). ASD is then defined as the average of all stored distances, which is 0 for a perfect segmentation.

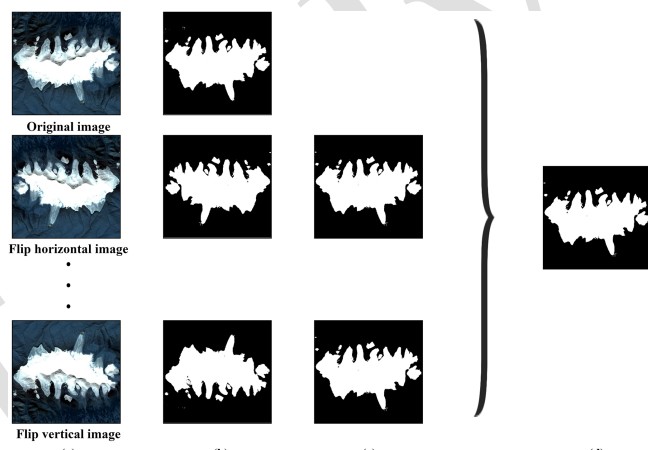

**Figure 8.** Test-time augmentation result. **(a–d)** RGB image obtained on 29 September 2020, its copies, prediction results, reductive copy results and the result of the vote, respectively.

The formula is given as follows:

$$B_{\text{seg}} = \{\forall p_1 \in A_{\text{seg}}, \text{closet\_distance}(p_1, p_2) | \exists \, p_2 \in A_{\text{gt}}\} , \tag{14}$$

$$\text{ASD} = \text{mean}(\{B_{\text{seg}}, B_{\text{gt}}\}) , \tag{15}$$

where $A_{\text{seg}}$ stands for the pixel of the boundary in the prediction result, and the same can be obtained for the definition of $A_{\text{gt}}$. $B_{\text{seg}}$ stands for the distance from the prediction result to gt, and the same can be obtained for the definition of $B_{\text{gt}}$.

## 3 Experimental result

This section will test the algorithms in this paper, analyze the experimental results and perform the following two tasks: (1) examine the performance of the Attention DeepLab V3+ with TTA on the test set, and (2) evaluate the ability of our method to extract large-scale glaciers.

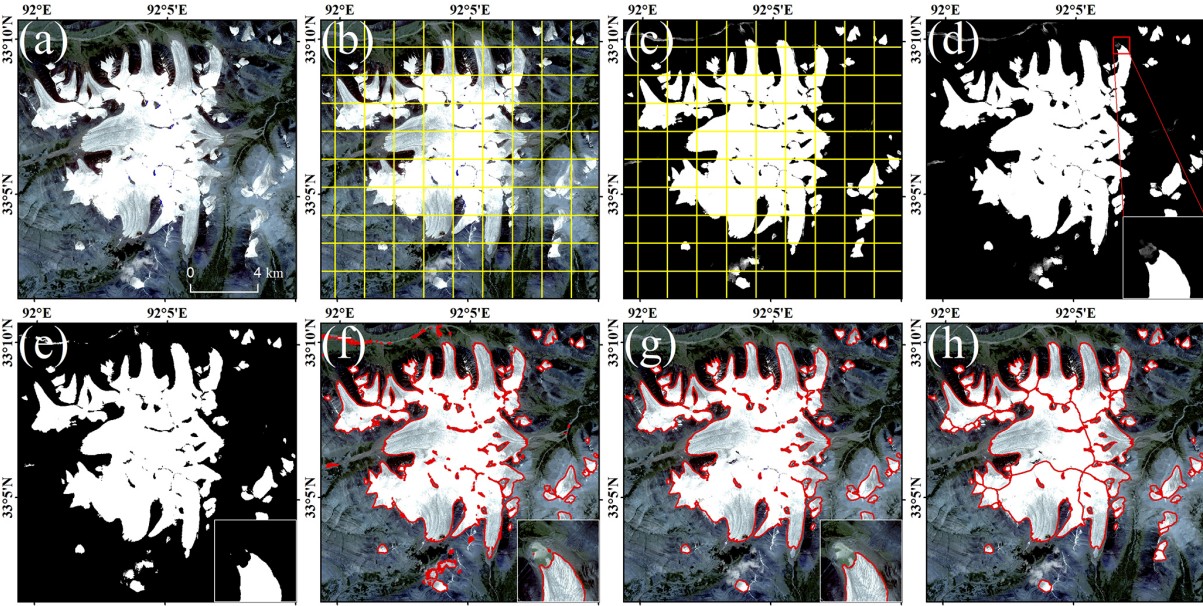

**Figure 9.** The procedure of glacier extraction. Firstly, the preprocessed image obtained on 19 August 2020. Panel **(a)** was clipped into $1024 \times 1024$ overlapping patches **(b)**, and then inputted into the model to obtain the predicted result **(c)**. Secondly, the results were merged, and pixel values greater than 0.5 were classified as glaciers to obtain the final binary map **(d** and **e)**. Thirdly, the binary map from above can be converted to a vector **(f)** and smoothed **(g)**. Finally, the glacier was segmented using ASTER GDEM (global digital elevation model) to obtain individual glacier vector boundaries **(h)**.

## 3.1 Experimental setup

For the experimental platform, we used a central processing unit with an Intel Core i9-10920 (3.50 GHz) processor, configured with 64 GB of memory, a Nvidia GeForce RTX 2080 Ti graphics card with 11 GB of video memory, the Windows 10 64 bit operating system and the Python programming implementation. In terms of software environment, we chose PyTorch as the deep-learning framework, CUDA version 11.1 as the graphics processing unit (GPU) computing platform and cuDNN8.0 as the deep-learning GPU acceleration library.

In this paper, the ratio of training set, validation set and test set was $8 : 1 : 1$. The training set was used to optimize the network parameters (weights and bias), the validation set to prevent overfitting and optimize the hyperparameters of the network (learning rate), and the test set to evaluate the effectiveness of the model trained on the training set, in which the images in the test set were not processed by data augmentation.

Gaofen-6 PMS images, with a spatial resolution of 2 m, allow for a large number of pixels occupied by a glacier. Therefore, 10 scenes of Gaofen-6 PMS images covering glaciers in regions of the Tanggula Mountains, the Kunlun Mountains and the Qilian Mountains were selected to test the capability of extracting large-scale glaciers by the model in this paper. The test images were acquired from July to December (Ta-

ble S2 in the Supplement). The whole visual procedure to extract glaciers is shown in Fig. 9.

## 3.2 Experiments on the Gaofen-6 test set

In this section, we explored the effectiveness of our network on the test set by comparing the results with FCN, U-Net and DeepLab V3+, in which the FCN that was specifically used was the FCN 32s network, and the backbone of U-Net was ResNet-18. Figure 10 shows the visualized prediction results of some test sets using different methods. The extracted results derived from FCN were almost error-free; however, it exhibited poor performance with the test set, which may be ascribed to its direct upsampling of 16, resulting in the loss of detailed information on glacier boundaries. The difference between the performances of U-Net and DeepLab V3+ on the test set was small. U-Net worked better in Fig. 10 rows 4, 5 and 6 and DeepLab V3+ in Fig. 10 rows 1, 2 and 3, with both methods occasionally misidentifying other features as glaciers. It is obvious that the Attention DeepLab V3+ with TTA model possesses the best glacier extraction capability, extracting glacier boundaries with excellent continuity and fewer fine patches.

Figure 11 shows the confusion matrix, kappa, MIoU, $F1$ score and ASD calculated for each model on the test set. In terms of kappa, MIoU and $F1$, the FCN network has the lowest accuracy of 0.9706, 0.9710 and 0.9838, respectively, because it cannot obtain the exact boundary of the glaciers. However, it has the smallest ASD error of

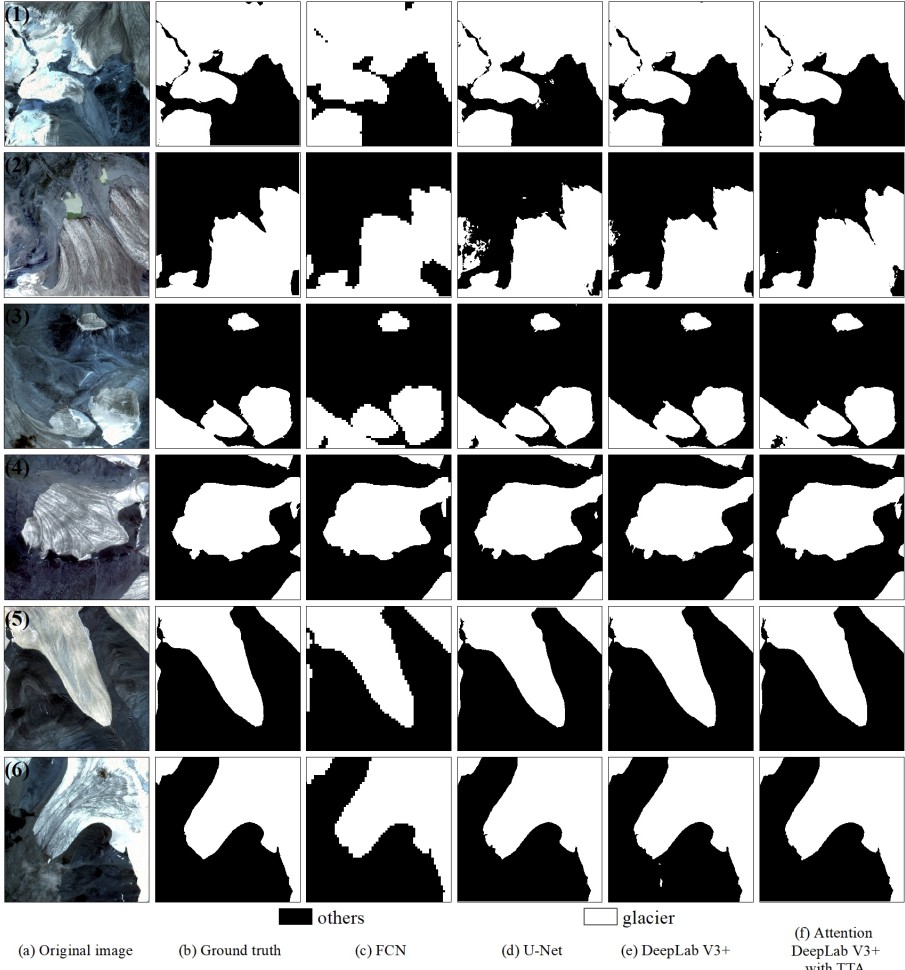

(a) Original image    (b) Ground truth    (c) FCN    (d) U-Net    (e) DeepLab V3+    (f) Attention DeepLab V3+ with TTA

■ others    □ glacier

**Figure 10.** Comparison of test results of different networks.

13.1686 px (26.3372 m). Since the test set has only a small range of images of $1024 \times 1024$ px, this network produces fewer fragment patches when classifying the test set, but it tends to ignore smaller glaciers when recognizing large-range images. Kappa, MIoU and $F1$ of U-Net were higher than FCN, and this method can obtain more accurate glacier boundaries. The ASD, however, is the highest (17.5268 px, 35.0536 m), which is attributable to its tendency to misidentify features, and the percentage of misidentified pixels is 0.60 %. The next is DeepLab V3+, which has an ASD error of 16.4501 px (32.9002 m) and a percentage of incorrectly identified pixels of 0.53 %. It also has higher kappa, MIoU and $F1$ accuracy compared to the first two methods. The Attention DeepLab V3+ with TTA model has the highest kappa, MIoU and $F1$, and the ASD is higher than FCN but lower than U-Net and DeepLab V3+. Moreover, the percentage of incorrectly identified pixels to total pixels is 0.42 % compared to DeepLab V3+, which is 0.11 % lower than DeepLab V3+. From the above analysis, it is shown that the capability of our network model is relatively high.

## 3.3 Experiments on Gaofen-6 images

In this section, we compared the ability of extracting complete glaciers based on 10 scenes of Gaofen-6 images (Sect. 3.1) using our method, the single-band threshold method (SBTM) and random forest (RF). Meanwhile, FCN, U-Net and DeepLab V3+ were selected for comparison because the complexity of the large-scale remote-sensing images may result in the performance with the test set being unsuitable for extracting huge complete glaciers. Some experimental results of each method are presented in Figs. 12–15, and kappa, MIoU, $F1$ and ASD were calculated for the extraction results to evaluate the performance of different methods (Table 1 shows the total accuracy, as detailed in Supplement Table S2).

When extracting complete glaciers, SBTM perform well when the spectral features of images are simple (Fig. 12) but has the worst effectiveness when the reflectance of some glaciers is similar to that of other features (Figs. 12 to 15). RF has better glacier extraction than SBTM due to its use of four

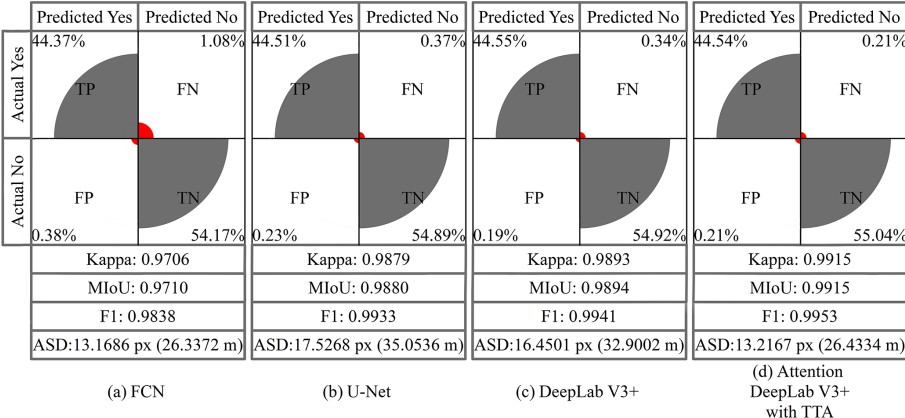

**Figure 11.** Performance of extraction models by confusion matrix, kappa, MIoU, $F1$ and ASD.

**Table 1.** Comparison of different glacier extraction methods on Gaofen-6 images.

| Evaluation metrics | SBTM | RF | FCN | U-Net | DeepLab V3+ | Attention DeepLab V3+ with TTA |
|---|---|---|---|---|---|---|
| Kappa | 0.9147 | 0.9555 | 0.9667 | 0.9745 | 0.9787 | 0.9818 |
| MIoU | 0.9217 | 0.9572 | 0.9675 | 0.9751 | 0.9791 | 0.9821 |
| $F1$ | 0.9316 | 0.9645 | 0.9736 | 0.9797 | 0.9830 | 0.9854 |
| ASD | 116.8237 px (233.6473 m) | 70.5443 px (141.0886 m) | 27.1095 px (54.2190 m) | 61.7339 px (123.4678 m) | 40.1783 px (80.3566 m) | 21.5695 px (43.1390 m) |

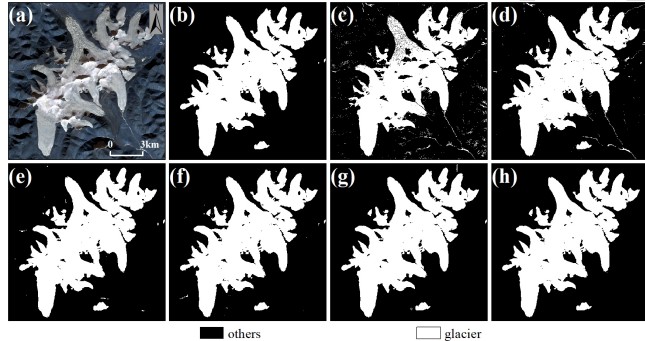

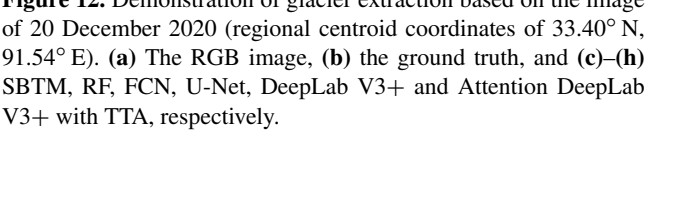

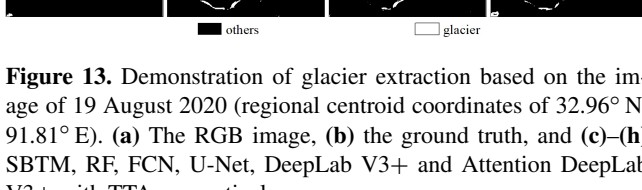

**Figure 12.** Demonstration of glacier extraction based on the image of 20 December 2020 (regional centroid coordinates of 33.40° N, 91.54° E). **(a)** The RGB image, **(b)** the ground truth, and **(c)**–**(h)** SBTM, RF, FCN, U-Net, DeepLab V3+ and Attention DeepLab V3+ with TTA, respectively.

**Figure 13.** Demonstration of glacier extraction based on the image of 19 August 2020 (regional centroid coordinates of 32.96° N, 91.81° E). **(a)** The RGB image, **(b)** the ground truth, and **(c)**–**(h)** SBTM, RF, FCN, U-Net, DeepLab V3+ and Attention DeepLab V3+ with TTA, respectively.

bands, but the method is prone to misidentify spectrally similar features, such as terminal moraine lakes. The deep learning yielded a better result than the above methods, which was trace-free and smooth despite the merging of many sample-sized images. Among the deep-learning methods, our method achieved the best performance, with extraction results similar to the ground truth.

For the three metrics kappa, MIoU and F1, SBTM has the lowest accuracy with 0.9147, 0.9217 and 0.9316, respectively, followed by RF and FCN, which obtained glacier boundaries that are rough. This was followed by U-Net and DeepLab V3+; they also achieve good results, obtaining more accurate glacier boundaries and extracting them better in the face of complex spectral features. Our method has

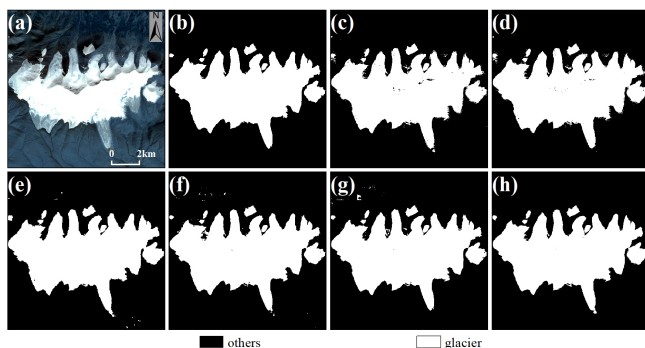

Figure 14. Demonstration of glacier extraction based on the image of 29 September 2020 (regional centroid coordinates of 35.83° N, 91.95° E). (a) The RGB image, (b) the ground truth, and (c)–(h) SBTM, RF, FCN, U-Net, DeepLab V3+ and Attention DeepLab V3+ with TTA, respectively.

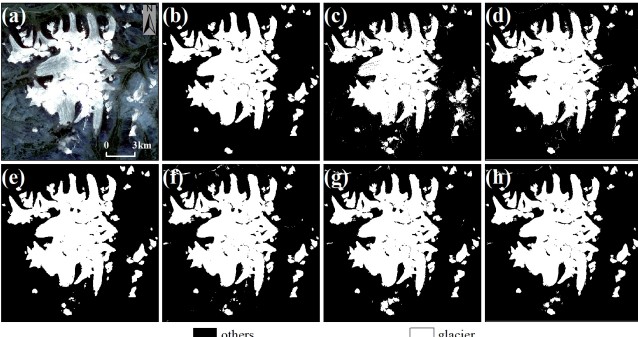

Figure 15. Demonstration of glacier extraction based on the image of 19 August 2020 (regional centroid coordinates of 33.11° N, 92.08° E). (a) The RGB image, (b) the ground truth, and (c)–(h) SBTM, RF, FCN, U-Net, DeepLab V3+ and Attention DeepLab V3+ with TTA, respectively.

the highest accuracy in the test images with 0.9818, 0.9821 and 0.9854 for kappa, MIoU and $F1$, respectively.

For ASD, SBTM also had the highest value of 116.8237 px (233.6474 m), and RF had the second highest value of 70.5443 px (141.0886 m). Both methods produced many fine patches, resulting in larger ASD. U-Net and DeepLab V3+ have smaller ASD compared to the previous two methods. Unlike its performance with the test set, FCN has more errors in extracting larger glaciers than our method and tends to ignore smaller glaciers due to direct upsampling, but it still has a smaller ASD of 27.1095 px (54.219 m). Our method produces a minimum ASD of 21.5695 px (43.139 m) when extracting a larger range of glaciers, which is 5.54 px (11.08 m) less than FCN.

## 3.4 Comparative experiment with or without TTA

We improved the accuracy of the Attention DeepLab V3+ model by employing TTA during testing and verified the ef-

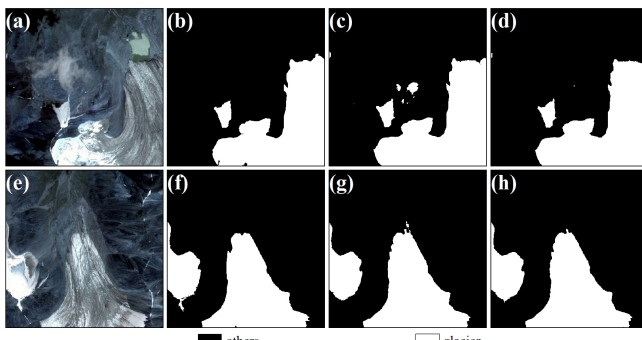

Figure 16. Examples of the results with and without TTA. Panels (a) and (e) are RGB images, (b) and (f) ground truth, (c) and (g) Attention DeepLab V3+, and (d) and (h) Attention DeepLab V3+ with TTA.

fectiveness by comparing the results with those without TTA. Table 2 shows the scores of each evaluation index, in which the Attention DeepLab V3+ is more accurate than the other networks tested in this paper with a higher kappa, MIoU and $F1$ (99.08 CE1, 0.9908 and 0.9948 TS1, respectively) than DeepLab V3+ (0.9893, 0.9894 and 0.9941, respectively), and the ASD was 2.5522 px (5.1044 m) lower. The addition of TTA increases the kappa, MIoU and $F1$ of the network by 0.0007, 0.0007 and 0.0005 TS2, respectively, and the ASD was 0.6812 px (1.3624 m) lower. This removes some discriminative errors of the pixels (Fig. 16), thus improving the performance of the model in extracting glaciers.

## 4 Discussion

### 4.1 Performance and wider application

The experimental results show that our method achieves advanced pixel-level classification performance in glacier extraction from high-resolution remote-sensing images when applied to 10 different regions of three different mountain ranges. The attention mechanism assists the network to increase the feature weights on glacial image elements, which makes the classification more accurate. In addition, the TTA strategy to increase network robustness makes it possible to achieve good results in different situations and can effectively distinguish glaciers from thin snow (Fig. 17a, b), thin clouds (Fig. 17c, d) and terminal moraine lakes (Fig. 17e, f). In the case of complex spectral features, it possesses a clear advantage in glacier extraction. Moreover, our method produces continuous glacier boundaries with reduced fine patches, which can markedly reduce the workload of further postprocessing.

However, when dealing with images with more snow or thicker clouds, it is difficult for the method in this paper to extract glaciers accurately because the real information of the surface is covered. Figure 18 shows high-resolution images

**Table 2.** Confusion matrix of test set results.

| Method | Attention DeepLab V3+ | | Attention DeepLab V3+ with TTA | |
|---|---|---|---|---|
| Pixel number | Predicted yes | Predicted no | Predicted yes | Predicted no |
| Actual yes | 18 6712 209 | 949 981 | 186 798 674 | 885 576 |
| Actual no | 966 562 | 230 801 649 | 880 097 | 230 866 053 |
| Kappa | 0.9908 | | 0.9915 | |
| MIoU | 0.9908 | | 0.9915 | |
| $F1$ | 0.9948 TS3 | | 0.9953 | |
| ASD | 13.8979 px (27.7958 m) | | 13.2167 px (26.4334 m) | |

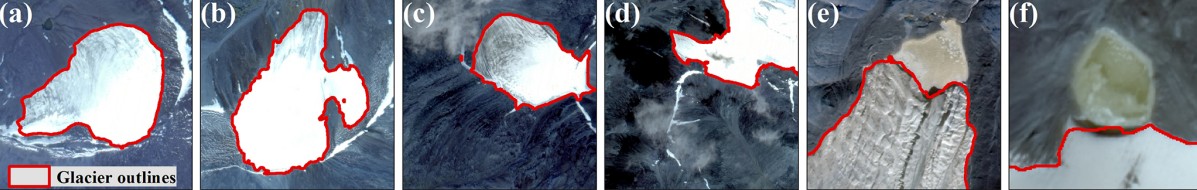

**Figure 17.** Examples of results in different situations. Panels **(a)** and **(b)** are the thin snow-covered, **(c)** and **(d)** the cloud-covered, and **(e)** and **(f)** the terminal moraine lakes.

of the same area in different periods and the extraction results of our method. The extraction is better when there is less snow in the image, and conversely it is easy to identify the snow as a glacier, which is a challenging problem to be solved when using optical remote-sensing images.

## 4.2 Advantages and limitations

Compared with previous studies on glacier extraction, the main achievements of this paper are three-fold: (1) building a glacier dataset with high spatial resolution for deep learning, (2) improving the DeepLab V3+ model by adding an attention mechanism, and (3) proposing an effective method to extract glaciers from high-resolution remote-sensing images.

We anticipated that our method differs from previous glacier extraction methods that focused only on a small scale, such as monitoring changes in the Antarctic ice shelf front and melting of the Greenland outlet glaciers (Baumhoer et al., 2019; Zhang et al., 2019), but it can be applied to extract a complete glacier over a large area. Unlike previous studies that relied on spectral features or information, such as texture and shape (Cheng et al., 2021; Zhang et al., 2021), high-spatial-resolution images possess more rich information of texture, shape and spatial distribution of ground objects, which contribute significantly to distinguish categories with similar spectral characteristics (Tong et al., 2020). Furthermore, a better glacier extraction method is expected to result in a more accurate glacier inventory.

Although our method does not work well when images are covered by large amounts of clouds or snow, we do not recommend using this method with poor quality data, considering that in most glacier inventory or glacier change research,

the images used are good quality images with fewer clouds and less snow. The method in this paper was not used to extract debris-covered glaciers, in which cases errors may occur. Additionally, certain features, such as frozen rivers, were sometimes mistaken for glaciers. Another drawback is that, despite the short prediction time of the well-trained model, its production was time-intensive due to the lack of readily available glacier datasets based on high-resolution remote-sensing images.

## 4.3 The difference between inventories

The amount of glacier resources is critical to regional water resources and future sea level rise (Bolch et al., 2012). In addition, the accurate extraction of glaciers contributes to the exact assessment of ice volume and mass balance, as well as the measurement of glacier length (Immerzeel et al., 2010). Existing glacier inventories are generally based on images with low to medium spatial resolution, which may misestimate global glacier size to a certain extent. Therefore, we discussed the differences between GGI and TPG2017 from the test images, in which the former was produced by manual delineation using Landsat Thematic Mapper (TM) and Enhanced Thematic Mapper Plus (ETM+) images in 1999–2003 (Nuimura et al., 2015), and the latter was generated based on Landsat Operational Land Imager (OLI) images in 2013–2018. To better explore the differences between the inventories, glaciers were firstly divided into accumulation zones and ablation areas based on the median area elevation that is deemed to be the material equilibrium line altitude (ELA), which is higher than the actual ELA for some glaciers

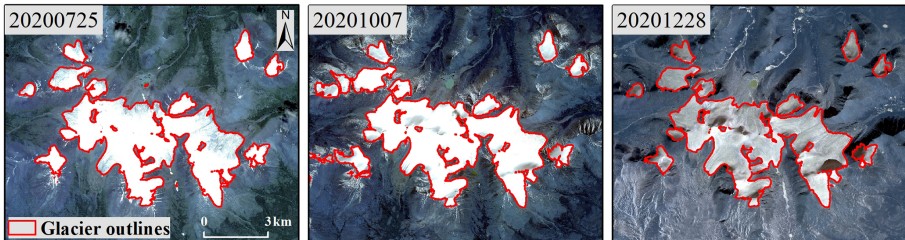

**Figure 18.** Examples of the results obtained by our method at different times in the same region (regional centroid coordinates of 32.96° N, 92.27° E).

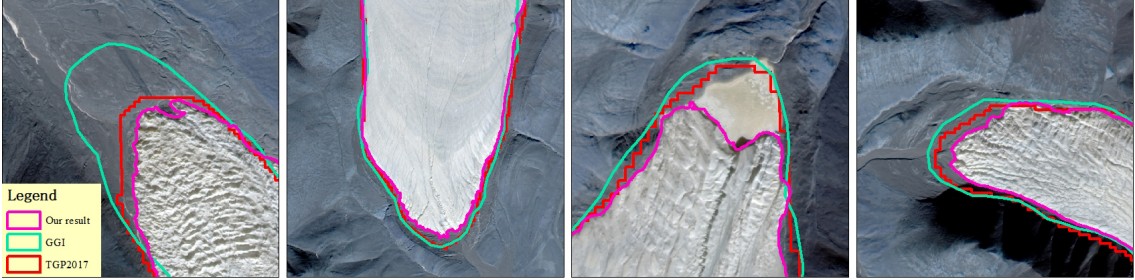

**Figure 19.** Comparison of different glacier inventories. Our data were obtained in 2020, the GGI was produced in 2002, and the TGP2017 was produced in 2017.

(e.g., disintegrating glaciers) (Braithwaite and Raper, 2009) but has little effect in this study (Table 3).

Our extracted glacier area differs from that of GGI by only $1.19\,\mathrm{km}^2$, accounting for $0.27\%$ of the extracted area, whereas the area difference with TPG2017 is $-21.02\,\mathrm{km}^2$, accounting for $-4.72\%$ of the extracted area (Table 3). Our extracted glacier boundaries have a high similarity in shape to GGI and TPG2017 (Fig. 19). Specifically, our model extracts a larger glacier area in the accumulation zone than GGI, which is mainly attributable to the omission of the shaded area in the upper glacier by GGI (Nuimura et al., 2015). However, the glacier area in the ablation zone obtained by our model is smaller than that in GGI, which is reasonable because of the retreat of glaciers in the ablation zone under dramatic warming (Fig. 20a–d). Compared with the results of TGP2017, the glacier areas extracted by our method are smaller in both the ablation and accumulation zones, while the difference is significantly larger in the ablation zone than in the accumulation zone, indicating that the glaciers' change is mainly concentrated in the ablation area, which is consistent with the general pattern of glacier changes (Fig. 20e–h). From the perspective of data sources, GGI and TPG2017 were produced by Landsat TM images with a resolution of 30 m and ETM+ and OLI images with a resolution of 15 m. Our data were obtained from Gaofen-6 PMS images with a resolution of 2 m such that fewer mixed pixels in the image allow for more detailed classification of glaciers and other features, as well as more accurate extraction of glacier boundaries, which leads to the smaller

glacier area for the data in this paper (Fig. 19). In general, our method allows more accurate extraction of the clean (debris-free) glaciers. Even though other features are misclassified as glaciers in a few cases, a small amount of manual modification can provide more exact data for glacier inventory.

## 5 Conclusions and prospects

In this paper, a glacier extraction method using the DeepLab V3+ network with attention mechanism and TTA was proposed to accurately extract glaciers from high-resolution remote-sensing images. This method can assist in improving the accuracy of automatically extracted glacier outlines and solving the problem of most high-resolution images not being able to extract glacier profiles using traditional methods, such as NDSI, due to the lack of a shortwave infrared band. By comparison with FCN, U-Net and DeepLab V3+, the superior glacier extraction ability of our model was demonstrated.

A total of 10 scenes of Gaofen-6 PMS images with glaciers were selected to test the ability of extracting complete glaciers over a large area. A comparison with other glacier extracting methods shows that our model achieves the best performance and could distinguish glaciers from terminal moraine lakes, thin snow and thin clouds. Moreover, the glacier boundary obtained from our method was continuous with fewer fine patches, thus substantially reducing the workload for postprocessing. When comparing the glaciers extracted by our method with the GGI and TPG2017, it was

**Table 3.** Summary of glaciers in GGI, TPG2017 and our data.

| Mountains | Region | Attention DeepLab V3+ with TTA | GGI | | | TPG2017 | | |
|---|---|---|---|---|---|---|---|---|
| | | Area | Area | Difference | | Area | Difference | |
| | | km$^2$ | km$^2$ | km$^2$ | % | km$^2$ | km$^2$ | % |
| Tanggula | Ablation | 102.97 | 112.37 | −9.4 | −9.13 | 112.28 | −9.31 | −9.04 |
| | Accumulation | 101.94 | 90.88 | 11.06 | 10.85 | 107.92 | −5.98 | −5.87 |
| Kunlun | Ablation | 61.01 | 61.05 | −0.04 | −0.07 | 62.80 | −1.79 | −2.93 |
| | Accumulation | 60.73 | 57.10 | 3.63 | 5.98 | 59.91 | 0.82 | 1.35 |
| Qilian | Ablation | 59.61 | 65.55 | −5.94 | −9.96 | 63.76 | −4.15 | −7.78 |
| | Accumulation | 59.31 | 57.43 | 1.88 | 3.17 | 59.92 | −0.61 | −0.66 |
| | Total | 445.57 | 444.38 | 1.19 | 0.27 | 466.59 | −21.02 | −4.72 |

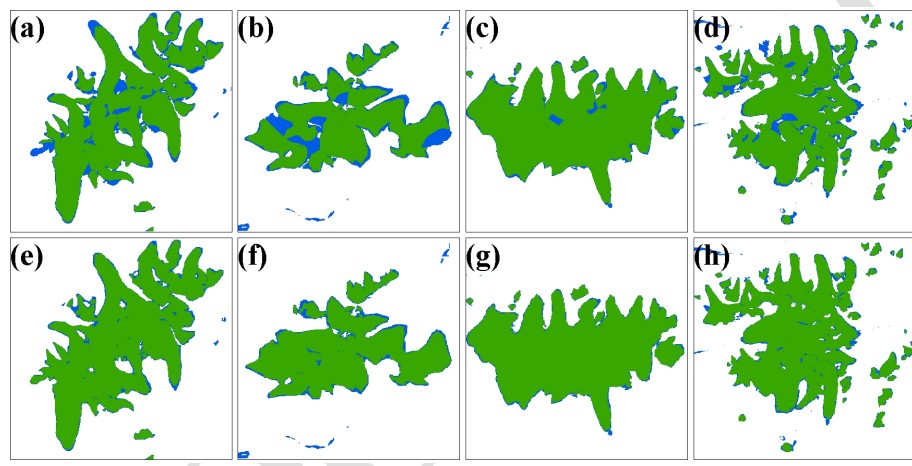

**Figure 20.** Panels **(a)**–**(d)** are the comparison between our results and the GAMDAM glacier inventory, and **(e)**–**(h)** are the comparison between our results and the TPG2017. Green and blue colors are the same and different areas in the two datasets, respectively.

found that our data have a more detailed representation of bare ice boundaries, which can provide more accurate data for glacier inventory after manual revision.

In the future, we will perform further research and adjustments in the following four aspects: (1) improving the algorithm to increase the network's ability to learn glacier features; (2) adding more samples to diversify the training samples and allow the network to learn more features to solve the existing difficulties of extracting debris-covered glaciers; (3) using other high-resolution remote-sensing images, synthetic aperture radar (SAR) images, etc., to compensate for the loss of extraction accuracy of optical images under cloud occlusion; and (4) applying transfer learning to reduce the time cost of sample annotation to allow deep learning to be applied to glacier extraction more quickly, thus improving model generalization.

*Code and data availability.* Gaofen-6 PMS images are available from China High-resolution Earth Observation System (https://www.cheosgrid.org.cn/, China High-resolution Earth Observation System, 2022). The TPG2017 is provided by National Tibetan Plateau Data Center (https://doi.org/10.11888/Glacio.tpdc.270924, Ye, 2019). The GGI used in this study is provided by https://doi.org/10.5194/tc-9-849-2015 (Nuimura et al., 2015). The sample datasets of glaciers and the code for the full deep-learning workflow are available from https://github.com/yiyou101/glacier-extraction.git (DOI: https://doi.org/10.5281/zenodo.7132888, Chu, 2022).

*Supplement.* The supplement related to this article is available online at: https://doi.org/10.5194/tc-16-1-2022-supplement.

*Author contributions.* XC designed this algorithm of extracting glacier outlines and wrote the original draft. XY contributed in terms of supervision and reviewing the manuscript. HD contributed

in terms of editing the manuscript. CC, JL and WP contributed in terms of getting satellite data and processing data.

*Competing interests.* The contact author has declared that none of the authors has any competing interests.

*Acknowledgements.* We thank the editors and the two reviewers for their valuable comments that improved the manuscript. We thank the China High-resolution Earth Observation System for providing high-resolution images used for this study.

*Financial support.* This research has been supported by the Project of China Geological Survey (grant no. DD20211570) and the National Natural Science Foundation of China (grant nos. 41861013 and 42071089).

*Review statement.* This paper was edited by Stef Lhermitte and reviewed by two anonymous referees.

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

**Remarks from the language copy-editor**

**Remarks from the typesetter**