# Peer review of "Glacier extraction based on high spatial resolution remote sensing images using a deep learning approach with attention mechanism"

_The Cryosphere, 2022_

## Author Response (AR1)

Response to referee #1

Dear referee,

We appreciate your comments and questions about our manuscript. All these comments are very important guides to improve the quality of our manuscripts. We will discuss and attempt to answer the points you raised in the following (replies are in blue):

General comments:

This manuscript describes the development of a deep learning methodology using neural networks and post processing of remote sensing imagery to automatically detect glacial boundaries. This work builds upon existing efforts in the field to address the labor-intensive task of surveying glacial changes over time for the purposes of further study. The method proposed uses an improved neural network design based on the Deeplabv3+ architecture to classify glacial pixels in the satellite imagery, and adds Convolutional Block Attention Modules to aid in identifying important glacier features during the automatic extraction task. The description of the method details is rigorous, which includes consideration such as the use of weighted Dice coefficient loss during neural network training, the use of image subsetting/patching to contend with computational limits/image constraints, and the use of Test Time Augmentation to produce a high confidence classification of glacial boundaries. The study evaluates this improved methodology on glaciers in the Tanggula and Kunlun Mountains using Gaogen-6 panchromatic/multispectral optical satellite imagery. The overall accuracy (99.58% out of 100%) and Kappa coefficient (0.9915 out of 1.0) of the resulting classifications is high, and while there are differences in the estimated glacial coverage between the method and existing glacial inventories, the study shows potential for further development and application.

Overall, the manuscript, its methodology, and its findings are sound. This study builds upon previous work, and it advances our understanding of automated glacial feature extraction, though it is limited in scope. There are some remarks to address and minor copy-editing which are listed below and under the specific comments. Given this, I recommend minor revisions with attention to comments, at the editor's discretion.

Thanks a lot for your approval.

Major comments:

1. One concern to raise is the choice of evaluation metrics. Existing literature in the field of automated glacial boundary extraction (Mohajerani et al. (2019, TC), Baumhoer et al. (2019, RS), Zhang et al. (2021, RSE), Cheng et al. (2021, TC), Robson et al. (2020), and He et al. (2021) use a wider variety of metrics not limited to OA and Kappa coefficients. More specifically, OA as a metric is subject to bias/skew depending on the test data, as small but important errors along glacial boundaries can be underrepresented given a large enough domain that is more easily classified. Most of the above also provide more robust accuracy metrics as the mean distance from the boundary in meters/pixels, and the Mean Intersection over Union. Thus, it is recommended that these two metrics be included for a more robust representation of errors along glacial boundaries, and allow for an easier comparison of this study's methodology with respect to others in the field.

Thanks for your valuable suggestion. We replaced the evaluation metrics with Kappa coefficient, Mean Intersection over Union, F1-score and average symmetric surface distance, which were used in the experimental results to compare our method with other methods in the field. However, we

calculated the average distance to the boundary in slightly different ways. For example, Marochov et al. (2021, TC), Baumhoer et al. (2019, RS), and Cheng et al. (2021, TC) studied the boundary of a specific glacier, such as its terminus or grounding line. Therefore, only the extracted boundary of the target glacier was used to calculate the average distance to the ground truth, which ignores errors of recognizing other features as glaciers in the imagery. Since the results of semantic segmentation are directly compared with the ground truth in our paper, the results obtained from semantic segmentation are usually the boundaries of all glaciers in the input image, so there may be cases where non-glacier regions are recognized as glacier when extracting boundaries. We take this into account and calculate the error of non-glacier region boundaries. The calculation formula for each metric is as follows.

The Kappa is a statistic that measures the agreement between prediction and ground truth, and can be calculated as follows:

$$p_e = \frac{(TP+FP) \times (TP+FN) + (FN+TN) \times (FP+TN)}{n^2} \tag{1}$$

$$kappa = \frac{\frac{(TP+TN)}{n} - p_e}{1 - p_e} \tag{2}$$

where $n$ is the number of total pixels; $TP$ and $TN$ are the number of pixels correctly predicted to be glaciers and correctly predicted to be other features, respectively; and $FP$ and $FN$ are the number of pixels incorrectly predicted as glaciers and incorrectly predicted as other features, respectively.

$MIoU$ for the binary classification problem is as follows:

$$MIoU = \frac{\frac{TP}{FP+TP+FN} + \frac{TN}{FP+TN+FN}}{2} \tag{3}$$

$F1$ is the harmonic average of precision rate ($P$) and recall rate ($R$):

$$P = \frac{TP}{TP+FP} \tag{4}$$

$$R = \frac{TP}{TP+FN} \tag{5}$$

$$F1 = 2 \times \frac{P \times R}{P+R} \tag{6}$$

$ASD$ is given in pixels/meters and based on the boundaries of the prediction result and ground truth (gt). For each pixel of boundary in the prediction result, the Euclidean distance to the closest pixel of another boundary in gt is calculated using the approximate nearest neighbor technique and stored. In order to provide symmetry, the same process is applied from the boundary of gt to the prediction result. $ASD$ is then defined as the average of all stored distances, which is 0 for a perfect segmentation. The formula is given as follows:

$$B_{seg} = \{\forall p_1 \in A_{seg}, closet\_distance(p_1, p_2) | \exists p_2 \in A_{gt}\} \tag{7}$$

$$ASD = mean(\{B_{seg}, B_{gt}\}) \tag{8}$$

where $A_{seg}$ stands for the pixel of the boundary in the prediction result, and the same can be obtained for the definition of $A_{gt}$. $B_{seg}$ stands for the distance from the prediction result to gt, the same can be obtained for the definition of $B_{gt}$.

2. Another concern is the scope/applicability of the technique to other glaciers, and under more difficult imaging conditions than just thin clouds/snow. While the methodology shows good results

on the testing data, it is limited to 4 ideal images of 2 locations (Tanggula and Kunlun Mountains). Existing works (Cheng et al., 2021) have proven the applicability of similar methods to wider applications, but additional testing on different domains/image conditions/SAR data may be useful ensure that this neural network has not overtrained on the training data. Since this may be outside the scope of the study (as touched upon in the conclusion), this may be done at the authors' discretion, but it may be of interest to see how well the neural network model has generalized from the data it has trained on.

Thanks for your helpful suggestions. As you mentioned, adding more test data would be very useful, but currently we have limited access to high-resolution images, so we can only test our method in as many cases as possible. To further demonstrate that our method can be applied to more areas, we added six control experiments in section 3.3. This includes 10 different areas in Tanggula Mountains, Kunlun Mountains and Qilian Mountains. In addition, we added a discussion section (Section 4.1) to discuss the effectiveness of this method under different conditions. Under more difficult imaging conditions where there is thick snow cover, the method may identify snow as glaciers, but we do not recommend the use of poor qualitied images, considering that glacier studies are usually based on good-quality images.

3. Section 4.2 provides a potentially valuable comparison between the study's glacial extent data product, and that of manually curated glacial extents. However, due to the time differences between the measurements (2020 for the study, 1999-2003 for GGI, and 2013-2018 for TPG2017), the comparison is not as useful as it could be. While this may also be out of scope for this study, it would be interesting to see a co-temporal comparison of glacial extents if such data exists. Otherwise, the differences in area/% changes make the potential errors from the methodology hard to separate from potential changes in the glaciers over time.

Thanks for your suggestion. As you said, it would make more sense to compare glacier datasets at the same year. However, the fact is that the Gaofen-6 satellite was launched on June 2, 2018, and the CGI and TPG2017 datasets were completed in earlier year. Undoubtedly, there are differences among these glacier datasets due to glacier recession. Considering that glacier change usually occurs in its ablation area, while its shape especially in accumulation zone remains stable. So the comparison in section 4.2 is to confirm that the data we acquired are similar in shape to those of GGI and TPG2017, proving the reliability of the data.

Specific Comments:
P2 L31: "researches of" -> "research on"
P2 L 37: "a relatively accurate results" -> "relatively accurate results"
P4 L82: "a upsampling" -> "an upsampling"
P4 L88: "internal correlation, its basic" -> "internal correlation. Its basic"
P18 L295: Zhang et al. (2019) and Cheng et al. (2021) both use single spectrum data inputs and don't utilize spectrum information, and already rely on texture/shape information.
P19 L305: "And," -> "Additionally,"
P20 L331: "2m that" -> "2m, such that", "more detail" -> "more detailed"
P20 L331- 335: Consider splitting these long sentences/rephrasing (i.e., "intact glaciers, although in a few cases…" -> "intact glaciers. However, in a few cases…"
P21 L347: "glacier extracted" -> "glacier extraction"

P21 L349: "extracting complete glacier" -> "extraction complete glaciers"

P21 L350: "And then comparison" -> "Comparison"

P21 L351: "which could distinguish glacier" -> "which could distinguish glaciers"

Thanks for your earnest suggestions. We corrected all these grammatic mistakes as you mentioned. With the help of Dr. Jake Carpenter whose native language is English, we carefully checked all sentences to avoid these stupid mistakes.

Response to referee #2

Dear referee,

We appreciate your comments and questions about our manuscript. All these comments are very important guides to improve the quality of our manuscripts. We will discuss and attempt to answer the points you raised in the following (replies are in blue):

General comments:

This paper explores the use of deep learning algorithms to map debris-free glaciers in Gaofen-6 PMS (pan/multispectral) imagery, which has 2-meter spatial resolution and lacks short-wave infrared (SWIR) bands. Previously employed glacier-mapping methods frequently rely on SWIR bands in instruments such as ASTER, Landsat, and Sentinel-2, because the reflectance of snow and ice is very low in that part of the spectrum. High-resolution (meter-scale) instruments tend not to have SWIR bands, as the authors point out, so if one needs to map (e.g. smaller) glaciers at high resolution, this method could be of great use.

Thanks a lot for your approval.

A limitation of this method is that it is designed to work on clean (debris-free) glaciers. Since debris cover is present on 44% of Earth's glaciers (https://www.nature.com/articles/s41561-020-0615-0), this method is currently limited to regional studies where glaciers are largely clean. However, the authors point out that in the future, the method could use more diverse input data and be made to work on debris-covered glaciers. The method described in this paper does appear to be a solid building block for future expansion.

Thanks for your valuable suggestions. Due to the region restriction of Gaofen-6 PMS imagery, we did not acquire those images where there are debris-covered glaciers, such as the Karakoram Range, Pamirs, southern Tien Shan, Himalayas, etc. Therefore, we did not test the debris-covered glaciers in this study because of the absence of sample. The differences between clean glaciers and debris-covered glaciers are relatively large, so it is essential to make a new sample of debris-covered glacier and to develop an optimized strategy. As you said, this method has a potential on debris-covered glaciers by expanding the dataset. I will be devoted to this solution in the coming time, which is also an important component of my dissertation.

While the authors provide a generally good overview of previous work, one closely related paper that was not cited is from Xie, et al. (2020), entitled "GlacierNet: A Deep-Learning Approach for Debris-Covered Glacier Mapping". The application space is a bit different for this approach, but this paper seems important to mention when discussing prior work.

Thanks for your valuable suggestion. This paper is cited in the revised manuscript.

This paper makes extensive use of deep-learning jargon, and it should be better described for this journal's audience. The first time the "attention mechanism" is mentioned in the main text (line 58), for example, it should be better introduced, or pointed out that it will be described in the next section. But even after that, the method is not adequately described. I think it would be good to briefly describe the concepts behind the jargon at first mention, for example, something like "ASPP is used to obtain multi-scale context information from the imagery." Otherwise, understanding of the gist of the article will be too reliant on the reader finding outside resources.

Thanks for your suggestions. We introduce some of the deep-learning jargon in more detail, such as DeepLab V3+, Convolutional Block Attention Module and Atrous Spatial Pyramid Pooling in the revised manuscript.

In summary, this paper describes a method that achieves good results in its currently limited domain, and appears to be a good building block for future extension of the method to other input data sets. I recommend publishing after significant revisions for 1) readability, and 2) adequate explanation of the concepts behind the algorithms.

Thanks a lot for your suggestion. For readability, we sought a native English-speaking scientific editor to polish our manuscript. For adequate explanation of the concepts behind the algorithms, we explain the deep-learning jargon in more detail.

Some specific comments:
I think for most people, longitude/latitude coordinates given in decimal degrees are more useful than degrees-minutes-seconds.

Thanks for your suggestion. We replaced the degrees-minutes-seconds to decimal degrees.

The paper would benefit from a go-through by a native (or near-native) speaker of English to correct usage of articles, plurals, etc.

Thanks for your suggestion. A native English-speaking scientific editor is polishing our manuscript to avoid these grammatic mistakes and inappropriate sentences.

---

## Referee Report (RR1)

**General Comments:**

The authors' response addresses the questions and comments raised by all reviewers, and integrates the feedback into the revised manuscript.

Specific revisions include the additional of robust evaluation metrics for comparing the neural network methodology to others in the field, and the inclusion of more testing/validation images to ensure the generalization/applicability of the method to other areas besides the primary study area. Furthermore, more text has been added describing the neural network methodology – while technical in nature, the text provides clarification of the work performed by the authors in this study. The authors acceptably address reviewer concerns, copyediting has been performed, figure captions have been clarified, and added/modified text is free from grammatical/syntactical errors.

After reviewal of the author's responses, as well as the changes to the revised manuscript, I can recommend that this submission should be accepted, at the editor's discretion.

**Specific Comments:**

N/A

---

## Author Response (AR2)

Dear Editor,

We appreciate your valuable comments on this manuscript, which are really important to us. As you pointed out, this manuscript currently has many shortcomings that need to be improved, and we have done our best to complete this work in this revision. The details are as follows:

Note: The black text is wordings in the original manuscript, the red text is the editor's opinions; the blue text is our reply, **and the bolded ones are wordings in the new manuscript.**

1. **Line 15:** pollin… polling

Reply: This is a spelling errors, thank you for the tip and we have corrected it.

2. **Line 120:** the hole… which hole? Not clear what is being meant here.

Reply: Thank you for your suggestion. The meaning of this is more difficult to understand. We corrected the sentence to "Dilated/atrous convolution allows us to explicitly control the resolution of features computed by deep convolutional neural networks and adjust filter's field-of-view in order to capture multi-scale information."

3. **Line 193:** approximate… what is approximate about it?

Reply: Thank you for your suggestion. Our expression is not quite accurate and 'approximate' has been removed.

4. **Line 451:** datasets including the GGI and TPG2017 used in this study are freely available.… Please indicate where the are available

Reply: Thank you for your suggestion. The download website of data sets has put on the revised manuscript.

5. **Line 454:** datasets of glacier can be provided upon request from the corresponding reader… It would be good practice to add a sample data set to the github page.

Reply: Thank you for your suggestion. We have added the datasets of glacier to the github page and the code for the full deep learning workflow are available from github.

6. i) clearly indicate where all data sets can be downloaded…

Reply: Thank you for your suggestion. The download website of data sets has put on the revised manuscript.

7. ii) include a sample dataset of glaciers in the github page + an example code that allows readers to effectively use your code.

Reply: Thank you for your suggestion. We have added the datasets of glacier to the github page and the code for the full deep learning workflow are available from github.